# Multiplicative Attacks with Essential Stealthiness in Sensor and Actuator Loops against Cyber-Physical Systems

**DOI:** 10.3390/s23041957

**Published:** 2023-02-09

**Authors:** Jingzhao Chen, Bin Liu, Tengfei Li, Yong Hu

**Affiliations:** 1Engineering Research Center of Metallurgical Automation and Measurement Technology, Ministry of Education, Wuhan University of Science and Technology, Wuhan 430081, China; 2School of Information Science and Engineering, Wuhan University of Science and Technology, Wuhan 430081, China; 3Science and Technology on Space Intelligent Control Laboratory, Beijing Institute of Control Engineering, Beijing 100190, China

**Keywords:** cyber-physical systems, cyber-attacks, false data injection, Kullback-Leibler divergence detector, Chi-square detector

## Abstract

Stealthy attacks in sensor and actuator loops are the research priorities in the security of cyber-physical systems. Existing attacks define the stealthiness conditions against the Chi-square or Kullback-Leibler divergence detectors and parameterize the attack model based on additive signals. Such conditions ignore the potential anomalies of the vulnerable outputs in the control layer, and the attack sequences need to be generated online, increasing the hardware and software costs. This paper investigates a type of multiplicative attack with essential stealthiness where the employed model is a novel form. The advantage is that the parameters can be designed in a constant form without having to be generated online. An essential stealthiness condition is proposed for the first time and complements the existing ones. Two sufficient conditions for the existence of constant attack matrices are given in the form of theorems, where two methods for decoupling the unknown variables are particularly considered. A quadruple-tank process, an experimental platform for attack and defense, is developed to verify the theoretical results. The experiments indicate that the proposed attack strategy can fulfill both the attack performance and stealthiness conditions.

## 1. Introduction

As the core and driving force of information technology in the past four decades, computers, communication, and control have caused a huge change in human social life. In this context, the concept of cyber-physical systems (CPSs) is proposed and widely valued. The CPSs involve most of the important industries that affect people’s livelihoods, making it important to study their key technologies [1].

The high integration of computing, physical plants, and communication networks increases the flexibility, reliability, and productivity of CPSs. However, it also extends the information security issues from the network layer to the computational and physical layers, resulting in serious safety problems [2]. Since the Stuxnet event in 2010 [3], more and more security incidents involving CPSs have appeared in the limelight. For example, the Flame virus suffered by the oil industry in the Middle East in 2012 [4], the advanced persistent threat cyber-attacks on a German steel company in 2014 [5], the Triton attacks on Schneider Triconex security instruments in 2017 [6], and the Watering Hole attacks on a municipal water treatment company in America in 2021 [7] can be easily listed. To address the security threats to CPSs, researchers focus on both attack and defense strategies. These studies are categorized into four types: attack methods [8,9], prevention of attacks [10], resilient control [11,12], and detection of attacks [13,14].

To cope with the potential stealth attacks in cyber-physical multi-agent systems, Nozari et al. [10] investigate the differential privacy issue of average consistency. The authors suggest that the differential privacy of communication information and the average consistency of multi-agents is a trade-off between them. To achieve differential privacy and thus prevent potential attacks, the agent states cannot be weakly converged to the exact average of their initial values. Abhinav et al. [11] studied a trust-based cooperative control strategy to counteract dual-channel false data injection (FDI) attacks in direct current (DC) microgrids. Sun et al. [12] suggest that a higher level of resilience can be obtained by improving the event-trigger mechanism to cope with denial-of-service attacks in networked control systems. The above studies consider the prevention and response to a single type of attack. For multi-class ones, such as stealthy FDI attacks and integrity and availability attacks, refs. [13] and [14] are two typical examples. Zhang et al. and Liu et al. propose a summation-based and a mode division-based detector, respectively. These methods are proven to be effective in the detection of stealthy attacks including, Stuxnet-like attacks.

In terms of attack strategies, an attacker attempts to deviate the actual output of the CPS from the defender’s desired reference trajectory. The extent of the deviation is termed the attack performance. The greater the deviation, the better the attack performance. However, the attacker cannot compromise a physical plant without recklessness. If a stealthiness condition cannot be satisfied, the attack will be detected, leading to its failure. Therefore, stealthiness is a necessary condition, and attack performance is the attacker’s goal. Existing work focuses on FDI attacks with particular stealth performance metrics, such as stealthy attacks [15,16,17], covert attacks [18,19], zero-dynamics attacks [20], optimal stealthy attacks [21], perfect stealthy attacks [22], and unpredictable attacks [23]. These attack strategies have a common design idea, that is, maximizing the attack impacts of the physical plant while ensuring the attacks cannot be detected by some specific detectors.

The Chi-square ($\chi^{2}$) detector is usually the object to be targeted when designing a stealthy FDI attack strategy. Mo et al. [24] propose a stealthy attack method in a linear CPS and analyze the relationship between the stability of the state matrix and the attackability of attack sequences. The authors also quantify the effects caused by the attacks in terms of the variation in the estimation error. In ref. [13], Ye et al. introduce two variable parameters $\left(\rho_{1},\rho_{2}\right)$ for the attack strategy in ref. [24] that can adjust the attack strength. After that, the authors further propose a self-generated approach in [8] for such kinds of attacks in refs. [13] and [24]. Since the $\chi^{2}$ detector only determines the anomalies of the state estimation residuals generated at the current detection instant, Mo et al. [25] present a method to construct a detector by measuring the difference of residuals in terms of Kullback-Leibler divergence (KLD). Li et al. [26] investigate a stealthy attack strategy acting in the actuator loop by solving an optimization problem to maximize the difference in residuals’ KLD before and after the attacks, under the constraint that the detection variable is lower than its threshold. Guo et al. [16] study a worst-case stealthy attack strategy in a remote state estimation scenario, where the KLD is chosen as the stealthiness metric. The authors present an innovation-based linear stealth attack, where the worst-case attack sequences are proved to obey a Gaussian distribution with a zero mean. In addition, the trade-off between the stealthiness of the attack and the impact on the system performance is discussed in [16]. Shang et al. [27] also investigate the worst-case stealthy attacks. The authors indicate the reasons for the method given in ref. [16] being a suboptimal solution and give an algorithm to obtain the optimal attack sequences without solving a semidefinite programming problem. Li et al. [21] present a stealthy attack strategy with an arbitrary mean Gaussian noise form, whose attack sequences are also obtained by solving an optimization problem.

All the above stealthy attack strategies against the KLD detector are designed in the context of remote state estimation scenarios for autonomous systems. In terms of attack strategies for generally controlled CPSs, Bai et al. [28] study an attack method acting in the actuator loop against the KLD detector with strict-stealthiness as well as a defined ε-stealthiness. Further, the solutions of optimal and suboptimal attack sequences are given for right-reversible and non-right-reversible systems, respectively. Furthermore, for the controlled CPSs, Ren et al. [9] investigate the stealthy KLD attacks operating in the sensor loop. The authors analyze the attack performance when the attack sequences satisfy the strict- and ε-stealthiness, as well as give the optimal solutions for the attack strategies.

It can be seen from the above that the existing attack strategies focus on the statistical properties of the state estimation residual z(k), and the attack sequences are in the form of time-varying additive signals generated online, whether for the $\chi^{2}$ or the KLD stealthiness. The residual z(k) can be expressed as $z(k)=y(k)-C\hat{x}(k)$, where y(k) denotes the output of the physical plant received by the control layer, $\hat{x}(k)$ stands for the estimation of the plant state x(k), and C refers to the output matrix of the physical plant. In practice, however, the process data supervision of CPSs is mainly dependent on the host PC’s configuration screen in the control layer. The unbiased statistical properties of z(k) cannot enable the anomaly-free display of the compromised sensor data y(k). See, for example, refs. [16,17]. This fact implies that the anomalies occurring in y(k) can lead to an immediate exposure of the so-called stealthy attacks. Therefore, making the compromised y(k) remain stealthy should be a necessary condition for all of the stealthy attacks, and it is obvious that the existing results ignore this critical issue. Moreover, the attack strategies with forms $y(k)=y_{p}(k)+y_{a}(k)$ and $u_{p}(k)=u(k)+u_{a}(k)$ depend on specific hardware and software environments to implement the real-time computation and data injection of the attack sequences in the sensor and actuator loops. These will undoubtedly increase the attack costs and the exposure risks. In the two formulas above, y(k) and u(k) denote the physical layer output received by the control layer and the control input to be sent to the physical layer, $y_{p}(k)$ and $u_{p}(k)$ represent the actual output and the received control input of the physical layer, respectively. The subscript “p” denotes “plant”. $y_{a}(k)$ and $u_{a}(k)$ refer to the attack signals injected by the attacker in the sensor and actuator loops, respectively, where the subscript “a” stands for “attack”.

Motivated by the above two concerns, this paper investigates a novel class of stealthy attack strategies with multiplicative constant attack matrix parameters. Two sufficient conditions for the existence of the attack matrices are given in the form of theorems. One of them is dominated by the stealthiness condition, and the other takes into account both attack performance and stealthiness. The contributions of this paper are as follows.
(1)A novel attack model with multiplicative attack matrices is proposed, which models the cyber-attacks as changes in the parameters of the generalized system consisting of physical and network layers instead of tampering with process data as given in existing literature.(2)The proposed attack matrices can be designed in a constant form, which can avoid the online operation of the attack parameters, thus reducing the costs and exposure risks of the attacks.(3)A definition of essential stealthiness is presented as a complement for the existing $\chi^{2}$ and KLD stealthiness conditions. Two sufficient conditions are given for the existence of constant attack matrices satisfying the essential stealthiness.(4)An attack and defense experimental platform is developed for CPSs, in which the physical plant is chosen as the classic quadruple-tank process. The platform can provide some experimental guidelines for related research in the field of CPSs security. The evaluations of the suggested strategies are performed in the form of hardware-in-the-loop experiments instead of simulations only, as in the existing work.


The subsequent sections of this paper are organized as follows. Section 2 introduces the basic model information of the CPS under consideration. Section 3 presents an attack model, gives a definition of essential stealthiness, and then analyzes the basic properties of the attack matrices. In Section 4, the methods for designing attack matrices are given in two theorems as the main results of this paper. In Section 5, the CPS attack and defense experimental platform are detailed, and the experimental results are analyzed, followed by conclusions and future work in Section 6.

## 2. Problem Formulation and Preliminaries

Consider the typical CPS illustrated in Figure 1, which consists of three parts: the control layer, the network layer, and the physical layer. The controlled plant lies in the physical layer, and its mathematical model can be represented as a discrete-time linear time-invariant (LTI) stochastic system:(1)xp(k+1)=Axp(k)+Bup(k)+wx(k),yp(k)=Cxp(k)+wy(k),
where xp(k)∈ℝnx, up(k)∈ℝnu, and yp(k)∈ℝny denote the system state, control input, and measurement output, respectively. The subscript ‘p’ denotes ‘plant’. A, B, and C are known model parameters, wx(k) and wy(k) are process and measurement noise, respectively.

Assume that the inherent controller of the CPS is employed in the control layer and has a form of dynamic output feedback:(2)xc(k+1)=Acxc(k)+Bcye(k)+Ecyr0,u(k)=Ccxc(k)+Dcye(k)+Fcyr0,
where xc(k) is the controller’s state, Ac, Bc, ..., Fc are the known controller parameters, where the subscript ‘c’ means ‘controller’. ye(k):=y(k)−yr(k) denotes the output error in the control layer, yr(k)∈ℝny and yr0 denote the output reference trajectory and the desired output stabilized value, respectively.

The output reference $y_{r}(k)$ is given by
(3)$$\left\{\begin{array}{l} x_{r}(k+1)=A_{r}^{\prime }x_{r}^{\prime }(k)+B_{r}^{\prime }x_{r0}^{\prime },\\ y_{r}(k)=C_{r}^{\prime }x_{r}^{\prime }(k). \end{array}\right.$$
y(k) in $y_{e}(k)$ and u(k) in Equation (2) denote the physical layer output received in the control layer and the control input that is to be sent to the physical layer, respectively; see Figure 1. Note that the studied CPS is a tracking control system, and that in practice the desired output reference is not always at the modeling equilibrium point of the physical plant. Without loss of generality, we assume that the desired output stabilized value satisfies $y_{r0}\neq 0$.

Attacks on CPSs may either occur in the physical layer or in the network layer. See, for example, refs. [1] and [29]. Since the attack sequences acting in the former can always be equivalently represented by the attack signals in the latter, Figure 1 simply gives the general form of the attack acting in the network layer. In particular, equations $u(k)=u_{p}(k)$ and $y(k)=y_{p}(k)$ hold under the assumption of an ideal communication network when the CPS is not under attack [20].

## 3. Modeling the Attacks

The attacker attempts to drive the state of the actual physical plant into the insecure regions by tampering with $y_{p}(k)$ and u(k) which are exposed to the communication network while satisfying the predefined stealthiness conditions. In this respect, existing stealthy attack strategies are mostly based on the two-channel FDI attack model [30] to implement the attack parameters design:(4)$$\;\left\{\begin{array}{c} u_{p}(k)=u(k)+u_{a}(k),\\ y(k)=y_{p}(k)+y_{a}(k), \end{array}\right.$$
where $u_{a}(k)$ and $y_{a}(k)$ are the attack sequences to be determined.

Notice that the attack sequences are usually time-varying to satisfy the stealthiness condition. This implies that the attack strategies with the form of Equation (4) in practice need to rely on some hardware and software settings to enable the real-time computation and data injection of the attack parameters. The necessary computing environments and the frequent interaction of network data increase the costs of attacks and their exposure risks. To this end, a new attack model is considered below.

According to the assumption of the desired output steady state $y_{r0}\neq 0$ in the previous section, the physical layer process data satisfying $u_{p}(k)\neq 0$ and $y_{p}(k)\neq 0$. Then a novel attack model can be formulated as
(5)$$\left\{\begin{array}{l} u_{p}(k)=A_{a}(k)\;u(k),\\ y(k)=A_{s}(k)\;y_{p}(k), \end{array}\right.$$
where $A_{a}(k)$ and $A_{s}(k)$ are the multiplicative attack matrices in the actuator and sensor loops, respectively.

Compared with the FDI model (4), the advantage of Equation (5) is that $A_{a}(k)$ and $A_{s}(k)$ can be designed as constant matrices $A_{a}$ and $A_{s}$ to avoid the online computational requirements of the existing attack parameters. At the same time, the numerical degrees of freedom for the two matrices can satisfy the stealthy attack design. The attack strategies in the form of $A_{a}$ and $A_{s}$ will be detailed in the next section. Before that, we begin with the analysis of the basic properties of the attack matrices in Equation (5).

**Definition** **1.**
*(Essential stealthiness of attacks.) For a CPS consisting of Equations (1)–(3) and the compromised communication network, an attack is said to satisfy the essential stealthiness with parameter γ>0, if its attack sequences make*

(6)
$${\left\|y_{e}(k)\right\|}^{2}<\gamma^{2}{\left\|\omega (k)\right\|}^{2}$$

*holds, where $y_{e}(k)$ refers to the output tracking error shown in the control layer, and ω(k) denotes the augmented disturbance in the closed-loop CPS affected by the attack sequences.*


Under Definition 1, the attack matrices of a stealth attack strategy with the form of Equation (5) should fulfill the following proposition.

**Proposition** **1.**
*(Non-singular properties of attack matrices.) For a CPS consisting of Equations (1)–(3) and the compromised network, a necessary condition for any attack strategy represented by Equation (5) to satisfy the essential stealthiness is that the attack matrices $A_{a}(k)$ and $A_{s}(k)$ are full rank.*


**Proof** **of** **Proposition** **1.**Consider the contrapositive of Proposition 1. When $A_{a}(k)$ and $A_{s}(k)$ have at least one set of dissatisfied ranks, the output controllability of the generalized system consisting of Equations (1) and (5) is not guaranteed. Then, for any desired trajectory $y_{r}(k)$ and generalized system output y(k), we have the tracking error ye(k)→0. At this point, the considered attack strategy does not satisfy the essential stealthiness. The proposition is proved. □

**Remark** **1.**
*The essential stealthiness condition given in Definition 1 is a complement to the existing ones defined based on the $\chi^{2}$ and the KLD detectors. The declared Equation (6) is founded on the fact that the operating data in the control layer should not be significantly abnormal when any attack is performed on the CPS; otherwise, it can be intuitively detected by the defender, making it difficult to achieve the intended attack purpose. The condition above is an essential requirement for a successful attack strategy and is therefore referred to as the essential stealthiness.*


Attack performance refers to the extent of deviation that the actual output of the physical plant causes from the defender’s desired reference trajectory. It can be defined as $\kappa :=\left\|{\bar{y}}_{p}-y_{r0}\right\|$, where ${\bar{y}}_{p}$ denotes the steady-state value of $y_{p}(k)$ in Equation (1). In particular, if the attack matrices are constant, κ can be expressed as $\kappa =\left\|A_{s}^{-1}{\bar{y}}_{e}+\right.$
$\left.(A_{s}^{-1}-I)y_{r0}\right\|$, where ${\bar{y}}_{e}$ denotes the steady-state value of $y_{e}(k)$ in Equation (6). For attacks with essential stealthiness, ${\bar{y}}_{e}$ has a small value, and thus the attack performance κ is mainly determined by the attack matrix $A_{s}$ based on the assumption of $y_{r0}\neq 0$ in Equations (2) and (3).

## 4. Attack Matrices Design for Stealthy Attacks

Consider the attack model Equation (5) with its constant matrices $A_{a}$ and $A_{s}$. This section gives two sufficient conditions for the existence of these matrices satisfying the essential stealthiness in a theoretical form. In particular, attackers aim to design the two matrices such that the compromised CPS fulfills both of the following two conditions. First, the physical plant state $x_{p}(k)$ deviates from its operation and causes the actual tracking error of the physical layer satisfying $\kappa =\left\|{\bar{y}}_{p}-y_{r0}\right\|\neq 0$. Second, the attack sequences meet the essential stealthiness with parameter γ>0, i.e., Equation (6) holds.

For condition one, the attacker considers an auxiliary perturbation signal d on $x_{p}(k)$. It can be represented as
(7)$$d+{\hat{x}}_{p}(0)=0,$$
where ${\hat{x}}_{p}(0)$ denotes the attacker’s estimation of $x_{p}(k)$ before the attacks are initiated, i.e., in the instant k=0. For the sake of exposition, we define the instant k∈(−∞, 0], similar to ref. [25], as the regular operation stage of the CPS and assume that the intended attack strategy is launched from the instant k=1.

Define $x(k):=x_{p}(k)+d$ and rewrite Equation (1) as
(8)$$\left\{\begin{array}{l} x(k+1)=Ax(k)+Bu_{p}(k)+A_{d}d+w_{x}(k),\\ y_{p}(k)=Cx(k)-Cd+w_{y}(k), \end{array}\right.$$
where $A_{d}:=I-A$. The physical plant model (from $u_{p}(k)$ to $y_{p}(k)$) in the attackers’ perspective is obtained.

**Remark** **2.**
*Notice that the closed-loop CPS without attack is stable and satisfies a definite static-error-free tracking control performance, i.e., $y(0)=y_{p}(0)\to y_{r0}$ holds. Hence, we have $E\left[x(0)\right]$
=0 when Equation (7) holds. It means that the auxiliary perturbation d gives Equation (8) a zero initial state in the statistical sense. This condition is easily neglected in the subsequent derivations.*


To satisfy the stealthiness conditions, the attacker introduces the following control layer output reference model
(9)$$\left\{\begin{array}{l} x_{r}(k+1)=A_{r}x_{r}(k)+B_{r}x_{r0},\\ y_{r}(k)=C_{r}x_{r}(k). \end{array}\right.$$

This model has the same structure as the inherent output reference Equation (3) of the CPS, but $A_{r}$, $B_{r}$, and $C_{r}$ are determined by the attacker with $x_{r}(0)=0$. $x_{r0}$ needs to satisfy $C_{r}{\left(I-A_{r}\right)}^{-1}B_{r}x_{r0}=y_{r0}$ to ensure that the steady output of Equation (9) is equal to $y_{r0}$. Here, $y_{r0}\neq 0$ denotes the desired output of the CPS before the instant of attack initiation (i.e., k=0), and $y_{r0}$ is constant around the attacks.

The following lemmas are introduced before giving our theorems. In the derivations of theoretical results based on the Lyapunov stability theory, the coupling of pending variables will lead to the corresponding matrices without the form of linear matrix inequalities. The following Lemma 1 is an important result of eliminating coupling terms, which will be used in the proof of Theorem 1.

**Lemma** **1.**[31] *For matrices T, S, M, N with appropriate dimensions, and a non-zero scalar β, the inequality $T+S^{⊤}M^{⊤}+MS<0$ holds if the following inequality is true.*


(10)
$$\left[\begin{array}{cc} T & *\\ \beta M^{⊤}+NS & -\beta N-\beta N^{⊤} \end{array}\right]<0.$$


The Schur’s lemma can combine matrix polynomials into a global matrix for convenience in subsequent derivation, but this process will introduce matrix inverse terms. The following Lemma 2 is used to perform the reduction in the nonlinear matrix inverse terms, after the main matrix is left and right multiplied a full rank square matrix.

**Lemma** **2.**[32] *For matrices X, Y, and J>0 with appropriate dimensions, the following inequality is true.*


(11)
$$XY+Y^{⊤}X^{⊤}\leq XJX^{⊤}+Y^{⊤}J^{-1}Y.$$


Similar to Lemma 1, Lemma 3 below also deals with the coupling of the pending terms. Lemma 1 is more concise. However, Lemma 3 is a necessary and sufficient condition. It introduces an extra matrix F that can increase the degrees of freedom in solving linear matrix inequalities under some circumstances. Lemma 3 will be used in the subsequent derivation of Theorem 2.

**Lemma** **3.**[33] *For matrices F, S, M, N, T with appropriate dimensions, and a non-zero scalar β, the following two inequalities are equivalent.*


(12)
$$\left[\begin{array}{cc} T+FS+S^{⊤}F^{⊤} & *\\ \beta M^{⊤}-\beta F^{⊤}+NS & -\beta N-\beta N^{⊤} \end{array}\right]<0,$$



(13)
$$T+S^{⊤}M^{⊤}+MS<0.$$


**Theorem** **1.**
*For the CPS given by Equations (1)–(3), if there exist matrices U, V, G, $G_{i}$, $P_{i}=P_{i}^{⊤}>0$, i=1, …, 4, and a scalar β>0, such that*

(14)
$$\left[\begin{array}{cc} T_{1} & *\\ T_{2} & T_{3} \end{array}\right]<0,$$

*holds, the attack determined by
$\left(A_{a},\;A_{s},\;d,\;\gamma \right)$ satisfies the essential stealthiness with parameter
γ, i.e., the Equation (6) holds.*


**Remark** **3.**
*In the literature related to stealthy attacks, it is typically assumed that the attackers have the complete model information of the CPS. See, for example, refs. [9,16,17]. For the attacker, a dynamic model with the closed-loop CPS information along with the attack parameters is employed in Theorem 1 (see Equation (A5) in Appendix A). This enables the attack matrices design can be turned into an $H_{\infty }$ control problem. An unstable physical plant state can easily lead to significant anomalies in the sensor measurements and causes the attack to be detected by the defender. The attacker has to take into account that the physical layer state should be driven to a new steady state, regardless of whether this steady state is technologically safe.*


**Remark** **4.**
*When the augmented system is stable from the attacker’s perspective, one has
$A_{s}y_{p}(k)\to y_{r0}$, i.e., a new desired output
${y^{\prime }}_{r0}=$
$A_{s}^{-1}y_{r0}$ holds. Therefore, the physical sense of the sensor attack matrix
$A_{s}$ can be reinterpreted. The injected
$A_{s}$ can indirectly tamper with the desired steady output of the physical plant (from
$y_{r0}$ to
${y^{\prime }}_{r0}$), making it possible for the attacker to manipulate the actual output of the physical system by simulating the sensor dead zone faults.*


In practice, attackers usually consider driving the output of the actual physical plant to a particular level while satisfying the essential stealthiness. The following theorem gives an implementation of this practical attack strategy.

**Theorem** **2.***Consider a CPS given by Equations (1)–(3). For a given attack target
${y^{\prime }}_{r0}\neq 0$ for physical plant output
$y_{p}(k)$ and a stealthiness parameter
$\tilde{\gamma }=y_{e}^{\mathrm{th}}{\left\|\omega (k)\right\|}^{-1}$ with
k>0, where
$y_{e}^{\mathrm{th}}=\sup(\left\|y_{e}(k^{\prime })\right\|)$,
$k^{\prime }\in \left(-\infty ,0\right]$, if there exist matrices
F,
$\tilde{U}$,
$\tilde{V}$,
${\tilde{G}}_{i}$,
${\tilde{P}}_{i}={\tilde{P}}_{i}^{⊤}>0$,
i=1,…,4, and a scalar
$\tilde{\beta }>0$, such that*(15)$$\left[\begin{array}{cc} {\tilde{T}}_{1} & *\\ {\tilde{T}}_{2} & {\tilde{T}}_{3} \end{array}\right]<0,$$*holds, the attack determined by
$\left(A_{a},\;A_{s},\;d,\;\tilde{\gamma }\right)$ satisfies the essential stealthiness with parameter
$\tilde{\gamma }$, where *${\tilde{T}}_{1}$*,*${\tilde{T}}_{2}$*, and *${\tilde{T}}_{3}$*are denoted as*(16)$${\tilde{T}}_{1}=T_{1}^{\prime }+FS+S^{⊤}F^{⊤},$$(17)$${\tilde{T}}_{2}=T_{2}^{\prime }-\tilde{\beta }F^{⊤},$$(18)$${\tilde{T}}_{3}=-\tilde{\beta }\;\mathrm{diag}\{\tilde{U}+{\tilde{U}}^{⊤},\tilde{U}+{\tilde{U}}^{⊤}\},$$$T_{1}^{\prime }$*and*$T_{2}^{\prime }$*have the same structure as*$T_{1}$*and*$T_{2}$*in Equation (14), respectively, except for the different symbolic representations of*$\tilde{V}$*,*${\tilde{G}}_{i}$*, and*${\tilde{P}}_{i}$. $A_{a}$*and*$A_{s}$*can be obtained by*$A_{a}=$${\tilde{U}}^{-1}\tilde{V}$*and*$A_{s}=\mathrm{diag}\left\{y_{r0}(1)/{y^{\prime }}_{r0}(1),\;\ldots ,y_{r0}(n_{y})/{y^{\prime }}_{r0}(n_{y})\right\}$*, respectively.*

**Proof** **of** **Theorem** **2.**The proof can be followed by Lemma 3 and Theorem 1, so it is omitted here. □

**Remark** **5.***Different from Theorem 1, here the sensor attack matrix*$A_{s}$*is a predefined diagonal matrix based on the attack target*${y^{\prime }}_{r0}$*. Since the full-rank square matrix satisfying*$y_{r0}=A_{s}{y^{\prime }}_{r0}$*is not unique, the predefined diagonal form reduces the degree of freedom for the results. Therefore, in the derivation of the decoupling terms, Theorem 2 considers the inequalities given by Lemma 3. The conservation of the result is reduced by introducing the new pending matrix*F.

## 5. Hardware Experiments and Results Analysis

In this section, a CPS attack and defense experiment platform is developed by employing the classic quadruple-tank process as the physical plant. Furthermore, the effectiveness of the proposed attack strategy is verified and analyzed in a hardware-in-the-loop (HIL) experiment.

### 5.1. Configuration of the CPS Experiment Platform

The developed experimental platform is shown in Figure 2. The plant consists of four components: a quadruple-tank process (QTP) [34], a defender’s industrial PC (IPC) equipped with a data acquisition board, an attacker’s PC, and the private Ethernet communication network.

The QTP is a representative type of coupled system in the process industry, where the controlled outputs are the liquid levels of the two lower tanks ($h_{1}$ and $h_{2}$), and the inputs are the control voltages acting on the two pump drive units ($v_{1}$ and $v_{2}$). The modeling method of the QTP can be referred to in refs. [14,34], and the model parameters of our experimental platform are
$$A=\left[\begin{array}{cccc} 0.9918 & 0 & 0.0036 & 0\\ 0 & 0.9918 & 0 & 0.0038\\ 0 & 0 & 0.9963 & 0\\ 0 & 0 & 0 & 0.9961 \end{array}\right],\;B=\left[\begin{array}{cc} 0.0354 & 0\\ 0 & 0.0374\\ 0 & 0.0115\\ 0.0122 & 0 \end{array}\right],\;C=\left[\begin{array}{cccc} 1 & 0 & 0 & 0\\ 0 & 1 & 0 & 0 \end{array}\right],$$
where the sampling period is $T_{s}=0.1\;s$, and the equilibrium point is selected as ${\bar{h}}_{1}=19.2\;\mathrm{cm}$, ${\bar{h}}_{2}=19.5\;\mathrm{cm}$, ${\bar{h}}_{3}=11.2\;\mathrm{cm}$, ${\bar{h}}_{4}=10.1\;\mathrm{cm}$, ${\bar{v}}_{1}=6.1\;V$, ${\bar{v}}_{2}=6.3\;V$.

As the hardware component of the control layer in CPS, the defender’s IPC is connected to the QTP through a data acquisition board and a wiring terminal board. The detailed hardware configuration of the platform is given in Table 1 for readers to replicate. Specifically, the controlled plant is composed of four cylindrical tanks in acrylic and six hoses in silicone. The two pumps and their drivers in Table 1 are the actuator part of the physical plant. They change the water flow in the hose in response to the received control signal. Two gas pressure sensors with model number HEYO-24V916PWM are chosen for real-time liquid level measurement. The readers can easily see such sensors in a blood pressure meter. The data acquisition board and the wiring terminal board are used together to transfer the collected liquid level signals to Simulink in real-time, while the latter’s control signals are processed by digital-to-analog conversion and then the control voltage can be output to the pump drivers.

The software components are based on the Real-Time Windows Target environment of the Matlab/Simulink to realize the HIL control of the QTP, as shown in Figure 3. The parameters of the employed dynamic output feedback controller (2) are
$$\begin{array}{c} A_{c}=\left[\begin{array}{cccc} 0.4701 & -0.1972 & 0.2548 & 0.2608\\ -0.2046 & 0.3359 & -0.3612 & 0.3587\\ 0.0736 & -0.2557 & 0.3799 & -0.1067\\ 0.1093 & 0.3661 & 0.1772 & 0.9371 \end{array}\right],B_{c}=\left[\begin{array}{cc} 0.0867 & -0.0240\\ 0.1465 & -0.0121\\ -0.1462 & -0.0217\\ -0.0416 & -0.0614 \end{array}\right],C_{c}={\left[\begin{array}{cc} -0.3921 & -0.0333\\ -0.1629 & -0.1501\\ 0.1740 & 0.2754\\ 0.0628 & 0.0092 \end{array}\right]}^{⊤},\\ E_{c}=\left[\begin{array}{cc} -0.0295 & -0.2416\\ 2.9535 & -0.6866\\ -0.6632 & -0.1423\\ 5.1465 & -0.7280 \end{array}\right],D_{c}=\left[\begin{array}{cc} -0.6333 & 0.0003\\ 0.0015 & -0.5508 \end{array}\right],F_{c}=\left[\begin{array}{cc} 0.2442 & -0.2035\\ -0.0082 & 0.1244 \end{array}\right]. \end{array}$$


The desired steady state output in the control layer is preset to be $y_{r0}={\left[3,\;5\right]}^{⊤}$ (indicating the actual levels 22.2 cm and 24.5 cm) and remains constant during the attacks.

The attacker uses the user datagram protocol (UDP) to tamper with the process data of the CPS through the private communication network to achieve the attack goal. Another typical protocol is the transmission control protocol (TCP), and a platform based on this protocol can be found in ref. [15]. To reduce the hardware investment, the input and output signal acquisition part of the physical layer of the experimental platform is designed on the IPC, while its software is separated from the controller. This idea is consistent with the basic principle of CPS and is able to meet the demand for attack data injection.

### 5.2. Experimental Results and Analysis

The attacker sets the attack target value of $y_{p}(k)$ as ${y^{\prime }}_{r0}={\left[6,\;8.3\right]}^{⊤}$, which corresponds to the actual levels 25.2 cm and 27.8 cm. The sensor attack matrix can be determined according to Theorem 2 as follows:(19)$$A_{s}=\left[\begin{array}{cc} 0.5 & 0\\ 0 & 0.6 \end{array}\right].$$

In addition, the stealthiness parameter is set to γ=0.0162, and the auxiliary perturbation d is:(20)$$d={\left[\begin{array}{cccc} -3.00 & -5.00 & -3.26 & -0.96 \end{array}\right]}^{⊤}.$$

The parameters in the output reference (9) are set as follows:(21)$$A_{r}=\mathrm{diag}\left\{0.25,\;0.5\right\},\;B_{r}=C_{r}=I,\;x_{r0}={\left[\begin{array}{cc} 2.25 & 2.5 \end{array}\right]}^{⊤}.$$

A feasible actuator attack matrix $A_{a}$ is obtained according to Theorem 2 as follows:(22)$$A_{a}=\left[\begin{array}{cc} 1.127 & 0.667\\ 0.206 & 0.893 \end{array}\right].$$

The total running time of the experiment is set to 1000 s. The attacker implements the attack strategy with parameters (19) and (22) from the 750th second to the end. The experiment results are shown in Figure 4, Figure 5 and Figure 6. To emphasize the attack process, these figures omit the first 500 s of irrelevant data.

Figure 4 shows the variation curves of the actual output $y_{p}(k)$ in the physical layer for the CPS. It can be seen that the physical system operates in a static-error-free tracking mode before the attack is launched. Once the attack starts, the two liquid levels deviate from the previous output reference ${\left[3,\;5\right]}^{⊤}$, and then stabilize at the attacker’s targets 6 cm and 8.3 cm, respectively. Thus, the attack performance can be obtained as
(23)$$\kappa =\left\|{\left[6,\;8.3\right]}^{⊤}-\right.\left.{\left[3,\;5\right]}^{⊤}\right\|=4.46\;.$$

The facts indicate that the attacker achieves the desired attack effect.

It should be noted that the actual outputs of the physical plant under attack have larger steady-state errors compared with those not under attack. This is because the injection of the sensor attack matrix $A_{s}$ indirectly tampers with the input gains $B_{c}$ and $D_{c}$ in CPS’s inherent controller Equation (2), resulting in a change in the dynamic relationship between the nominal output u(k) and input $y_{e}(k)$ in the control layer. Moreover, the actuator attack matrix $A_{a}$ can be considered as the tampering of the control gain for the physical plant in Equation (1), that is, B is altered to $BA_{a}$. Almost all existing studies treat the attack process as the operating data being compromised. It is important to analyze both the attack and defense issues of CPSs from the perspective of model parameters under attack because the attack model essentially raises open questions.

In addition, Figure 5 and Figure 6 give the curves displayed in the control layer before and after the attacks. They present the output y(k), tracking error $\left\|y_{e}(k)\right\|$ (the norm form of $y_{e}(k)$), and the implicit metrics of tracking control performance $\gamma \left\|\omega (k)\right\|$ in the defender’s perspective. It can be seen that although the actual output of the physical plant deviates from the predefined reference when the CPS is subjected to attacks (as shown in Figure 4), the received data in the control layer are not significantly abnormal and only present a phenomenon similar to a short-time exogenous perturbation. From the perspective of the process data received in the control layer, the Equation (6) holds after a limited tuning time (about 29.1 s), which indicates that the designed attack matrices satisfy the desired stealthiness.

### 5.3. Discussions and Limitations

As we can see from the experiments in Section 5.2, the attacker can simply obtain the attack matrices in constant form by solving the linear matrix inequalities offline in a one-time process. Thanks to the constant form of the attack matrices, the attacker can, in practice, diagonalize the attack matrices and then implement the attack by adjusting the gain knobs of the sensors and actuators in the physical layer. This physical attack does not require any data injection in the communication network and is strictly stealthy for most anomaly detectors based on network traffic monitoring. In addition, the phenomenon, such as the short-time exogenous perturbation that appears in Figure 6, is essentially caused by the change in the potential desired trajectory due to the sensor attack matrix. This phenomenon can be weakened when the attacker tries to target the stepped desired trajectory as the attack goal.

It is important to note that the proposed attack strategies also have some limitations. On the one hand, we assume that the attacker has complete information on the CPS model. This assumption is common in the research of stealth attacks (e.g., in ref. [9,16,17]), but it is worth being concerned to weaken this assumption in order to further improve the practical value of the attack programs. This paper does not make a breakthrough in this regard. On the other hand, the two proposed theorems assume that the attacker tries to drive the output of the physical plant to an unsafe constant value. This linearized attack goal remains somewhat homogeneous due to the constraints of the stealthiness conditions. More flexible forms of attack targets should be considered in the future.

## 6. Conclusions and Future Work

In this paper, we studied a class of stealthy attack strategies in the form of multiplicative matrices against CPSs. Two theorems are given for designing attacks with constant matrices. The model of multiplicative attack matrices is proposed for the first time. The attack strategies based on this model transform the attack parameter designs into controller design issues, which may raise a new research trend in the field of stealthy attack strategies. Moreover, a CPS attack and defense experimental platform is developed. The platform has the characteristics of low cost and high openness, which promotes the practicalization of stealthy attack research.

In the future, our research will be extended to stealthy attack methods under the unknown model information. The emphasis should be on data-driven multiplicative stealthy attacks. In addition, the relationship between the proposed stealthiness conditions with the existing $\chi^{2}$ and KLD ones should also be investigated.

## Figures and Tables

**Figure 1 sensors-23-01957-f001:**
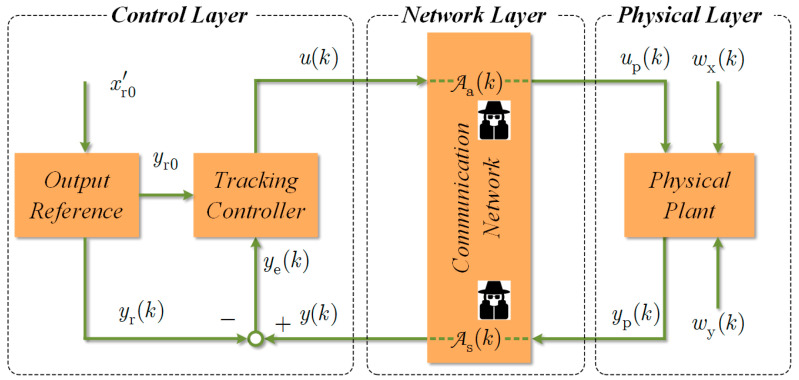
Block diagram of a CPS subject to attacks.

**Figure 2 sensors-23-01957-f002:**
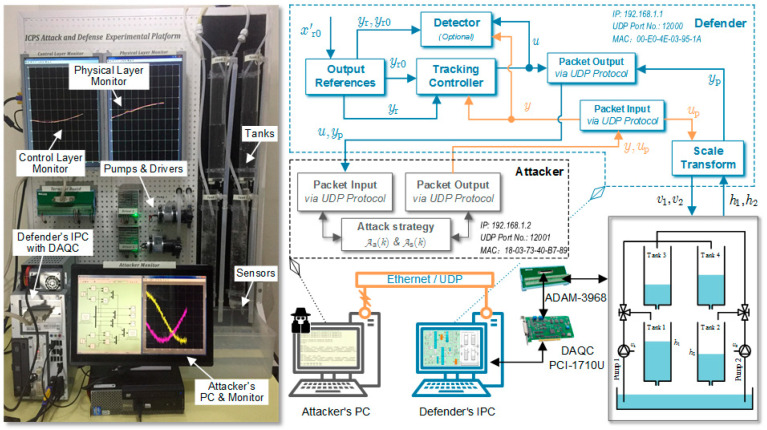
The structure of the experimental platform and a photo in operation.

**Figure 3 sensors-23-01957-f003:**
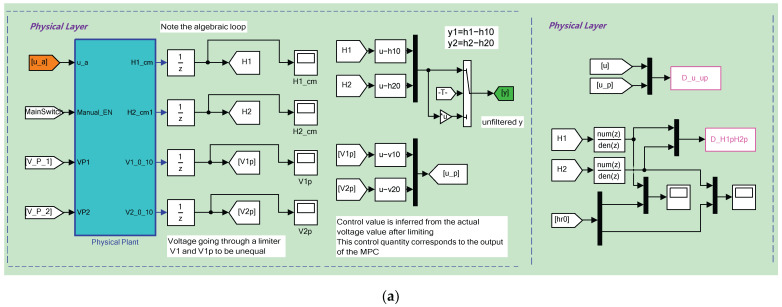
Software implements for HIL experiments in Matlab/Simulink: (**a**) The physical layer; (**b**) The first part of control layer; (**c**) The second part of control layer; (**d**) The network layer and attacks therein.

**Figure 4 sensors-23-01957-f004:**
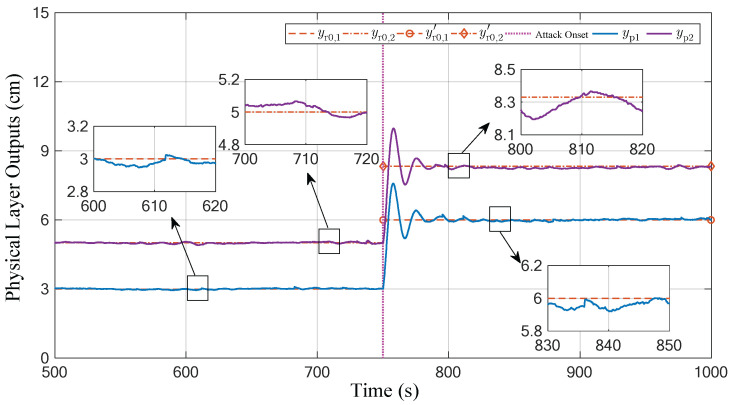
Actual output curves of the physical plant: the attack initiation time is 750 s, see the magenta vertical dotted line.

**Figure 5 sensors-23-01957-f005:**
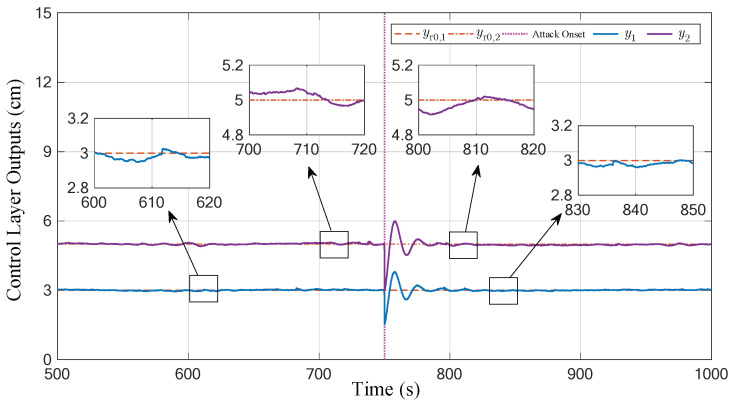
Output curves displayed in the control layer.

**Figure 6 sensors-23-01957-f006:**
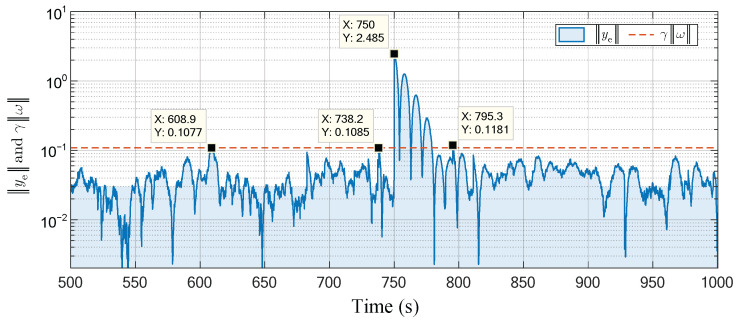
The curves of $\left\|y_{e}(k)\right\|$ and $\gamma \left\|\omega (k)\right\|$ before and after the attacks.

**Table 1 sensors-23-01957-t001:** Hardware configuration of the experimental system.

Components	Attribute	Description
Water Tanks	Quantity	4
	Material	Acrylic
	Depth	350 mm
	Inner Diameter	44 mm
Pumps	Quantity	2
	Type	Diaphragm
	Rated Voltage	24 V DC
	Load Current	0.8–1.5 A DC
	Rated Flow	2.2 L/min
	Head	10 m
Drivers	Quantity	2
	Model Number	HEYO-24V916PWM
	Control Voltage	0–10 V DC
	Output Voltage	0–24 V DC
	Static Current	0.01 A
Sensors	Quantity	2
	Model Number	XGZP6887
	Range	0–5 kPa
	Output Voltage	0.5–4.5 V DC
	Accuracy	1.5 %Span
	Compensation Temperature	0–60 °C
Data Acquisition Board	Producer	Advantech
	Model Number	PCI-1710U
	Input/Output Type	Analog
	Input Quantity	16
	Output Quantity	2
	Sampling Rate	Max. 100 kHz
	Accuracy	16 bit
	Others	Digital I/O & Counters
Wiring Terminal Board	Producer	Advantech
	Model Number	ADAM-3968
	General	68-pin/DIN-rail Mounting

## Data Availability

Not applicable.

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
