# Peer review of "Multiplicative Attacks with Essential Stealthiness in Sensor and Actuator Loops against Cyber-Physical Systems"

_sensors, 2023, doi:10.3390/s23041957_

Round 1
Reviewer 1 Report
This paper presents a proposed a multiplicative attack with essential stealthiness for sensor and actuator loops against cyber physical system.
- Through the paper, authors also claimed that the proposed essential stealthiness condition is the first time proposed and complements the existing ones. The experiments conducted through this study (a quadruple-tank process experimental platform) indicated that the proposed attack strategy can fulfill both the attack performance and the stealthiness conditions. I found the proposed algorithm has a significant contribution on its domain. The flow of analysis is performed also quite clear.
- To improve the significant of this paper, please find the following some of my comments and suggestion for author’s research work:
o I found this paper has too many formulas used and matrixes etc. Sometimes when I read, I loss maybe because the continuous used of formula-by-formula, theorem by theorem, lemma by lemma in too long paragraph. This part I suggest author to make some improvement in your writing regarding this scenario. I found it as a critical flaw of this paper.
o Result and discussion part need major improvement.
o Reference No 34, quite obsolete – year 2000, that was 22 years ago. I suggest author to find more current paper related.
o The abbreviations declared at the end of this paper looks a bit awkward. I think enough if author declare the full name when the terminology is first appeared in paper.
o Conclusion paper need improvement. It sounds like author just list down what you did, but analysis result should be describe more here to convince readers of your strong contribution.
o No future work mentioned for this paper. Important to add here.
o Figure 3, font used is extremely small and couldn’t be read properly. Increase the quality of that figure. Maybe divide into few figures (part by part) would be better.
o Discussion for Table 1 is too short. Need more explanation.
o Line 387 – “Figures 4-6” Figure 4 until Figure 6
o Line 42 – USA – full name required here, please check your entire paper and make sure any short form used/ abbreviations, please declare it full name when it first appear.
Reviewer 2 Report
The topic is suitable to the intended MDPI journal.
The authors provide a solid theoretical treatment, accompanied by experimentation results.
The article can be improved significantly if the authors:
- provide in the introduction information on more articles dealing with stealthy attacks
- explain what x, y, z, u, .... are before their first use (p. 2); same for the indices
- describe the structure of the article at the end of the introduction, ...
The article lacks a discussion of the results, e.g.:
- what are the limitations of the study (for example, the authors seem to assume that the attacker has a very good knowledge of the attacked system)
- the particular linear treatment of the attack performance (while in practice the attacker will aim to achieve some events not foreseen by the system designer, e.g. overflow of the tanks, or breakage of the centrifuges in the Stuxnet example)
- follow on studies, etc.
The language is very good. Yet, I'd advise the authors to consider replacing 'security prevention' with prevention of attacks, 'security detection' with detection of attacks (both on lines 44-45) and potentially add response to attacks (in addition to resiliency) and respective sources.
Round 2
Reviewer 1 Report
This updated version has been improved quite a lot by authors. I feel satisfied with all the corrections made by authors. I found all comments are being taken into consideration and improvement made increase the quality of this paper indeed.
Just maybe some update for the Appendix which need to be put after the References.
Other part is all fine as it is.